# The Effect of Feldspathic Thickness on Fluorescence of a Variety of Resin Cements and Flowable Composites

Joana Santos de Cunha Pereira [1,2,*], José Alexandre Reis [1,3], Francisco Martins [1,3], Paulo Maurício [1,3] and M. Victoria Fuentes [4]

1 Oral Rehabilitation Department, Instituto Universitário Egas Moniz, Monte de Caparica, 2829-511 Almada, Portugal; jreis@egasmoniz.edu.pt (J.A.R.); fmartins@egasmoniz.edu.pt (F.M.); pmauricio@egasmoniz.edu.pt (P.M.)
2 Faculty of Health Sciences, Rey Juan Carlos University, 28922 Madrid, Spain
3 CiiEM, Instituto Universitário Egas Moniz, Monte de Caparica, 2829-511 Almada, Portugal
4 Idibo Research Group, Area of Stomatology, Faculty of Health Sciences, Rey Juan Carlos University, 28922 Madrid, Spain; victoria.fuentes@urjc.es
* Correspondence: jpereira@egasmoniz.edu.pt; Tel.: +351-9-1629-0724

**Abstract:** (1) Background: The shade of resin-based materials and ceramic thickness influence the optical color of laminate restorations. The purpose of this study is to evaluate—in vitro—the effect of resin-based cement shade and ceramic thickness on fluorescence of feldspathic laminate veneers; (2) Methods: 180 samples of feldspathic ceramic A2 shade with two different thicknesses (0.5 and 0.8 mm) were obtained. The samples were cemented to composite resin substrates with one of the following materials in different shades (n = 10): resin cement (Variolink Esthetic in Light, Neutral and Warm shades; or RelyX Veneer in B0.5 /white, Translucent and A3 Opaque/yellow opaque shades); flowable composite resin (G-aenial Flo in A2 and A3 shades) or a pre-heated composite resin (Filtek Supreme XTE, A3 body shade). The fluorescence spectra were obtained by means of a spectrofluorometer. Two-way ANOVA, Tukey, and Student's $t$-tests were performed ($\alpha = 0.05$); (3) Results: Fluorescence values were significantly influenced by the resin-based agent tested ($p < 0.001$), the thickness of ceramic ($p < 0.001$), and their interaction ($p < 0.001$). The lowest fluorescence values were achieved by RelyX Veneer resin cement regardless its shade and the ceramic thickness; (4) Conclusions: both the shade of resin-based agent and the feldspathic ceramic thickness influenced the fluorescence of laminate restorations.

**Keywords:** feldspathic ceramic; resin cement; flowable resin; fluorescence; thickness





## 1. Introduction

Nowadays, ceramic laminate veneers are one of the main choices to perform a highly aesthetic oral rehabilitation [1–3]. Feldspathic porcelain, the first type of ceramic used in dentistry, is a suitable clinical solution for fabricating veneers, due to excellent esthetic and biocompatibility properties and long-lasting performance [1–4]. Long-term success of ceramic veneers depends partly on the adhesive cementation [4,5]. Once adhesively cemented, ceramic laminate veneers exhibit an increased fracture strength and propitious success rates [6–8].

The restoration's aesthetic goal should reproduce the optical characteristics of the natural tooth. Several factors—such as ceramic thickness, cement, and abutment color—influence the final color of the restored tooth [9–14]. Translucency, opalescence, and fluorescence are other optical properties that alter the overall appearance of the restoration [15].

Fluorescence of the natural teeth occurs when their surface absorbs ultraviolet light (UV) (350–400 nm) and emits light with a longer wavelength, creating a bluish-white color [16–21]. This property is mostly determined by dentin because of the greater amount of organic material, which contains fluorescence-releasing amino acids—such as tryptophan,

providing three times more fluorescence than enamel [11,20,22–24]. Fluorescence can be classified as distinctive clinical optical property that not only makes teeth appear whiter but also brighter by emitting more blue radiation due to fluorescence, which converts UV radiation (invisible to the eye) to blue radiation (visible to the eye). Thus, to provide better integration, it is mandatory that teeth restored with ceramic veneers have a fluorescence emission similar to that of the natural teeth [19].

The type and composition of the ceramic will influence the intensity of fluorescence [19]. Some luminescent additives—such as europium and other rare-earth elements that exhibit visible fluorescence—are included in composition of ceramics and resin cements to obtain fluorescence properties similar to the tooth structure [19,25–28]. To date, few studies have evaluated the fluorescence of ceramic restorations. Rafael et al. [27] evaluated the impact of tooth substrate shade on color differences, transmittance, and fluorescence of CAD-CAM (Computer Aided Design and Computer Aided Design Manufacturer) leucite based ceramics. The association of ceramic samples with darker substrates decreased fluorescence intensity. Silami et al. [28] showed that the apparent fluorescence of laminate veneers was influenced by the combination of two different ceramic veneers and the cement (light-cured or self-adhesive dual resin cements). Other authors also showed that high-fluorescence resin-based cement may interfere with the final esthetic result of thin restorations [20].

Light-cure resin cements are indicated when luting relatively thin and translucent restorations, as it allows light irradiance to activate the photo-initiators [29]. This type of resin cement exhibits clinical advantages such as long period of working time, setting on demand, and better color stability [30]. Recently, there has been an increasing trend to use flowable composite resins as light-cure cements for adhesive luting [29] in order to benefit from their physical properties (more filler-loaded than resin cements), as well as an improved cost–benefit compared to resin cements [31].

Several devices and methods for analysis of fluorescence in aesthetic materials have been employed in previous papers. Fluorometers or spectrofluorometers are commonly used because they provide quantitative results without the limitations of photography methods [19,27]. These devices measure fluorescence parameters, such as intensity and distribution, at various wavelengths. An emission spectrum corresponds to the wavelength intensity distribution of the emitted fluorescence at a constant excitation wavelength [19].

Due to higher translucency of feldspathic ceramic, the brand and shade of material used for the cementation may interfere with the fluorescence of the restoration. Therefore, the aim of this in vitro study was to evaluate the effect of light-cure resin cement and flowable composite resins in different shades on fluorescence of CAD-CAM feldspathic veneer restorations in two thicknesses. Our research hypothesis was that the emission intensity of fluorescence of feldspathic ceramics restorations is not influenced by the shade of resin-based material, nor by the thickness of feldspathic veneer.

## 2. Materials and Methods

The materials used in the present study are listed in Table 1.

Five CAD-CAM feldspathic ceramic (CEREC Blocs; Denstply Sirona, PA, USA), A2 shade, were used for the present study (10 × 12 mm). The ceramic blocks were cut into slices with thicknesses of 0.5 and 0.8 mm (ninety slices for each thickness) with a water-cooled diamond saw (Isomet 1000; Buehler, Lake Bluff, IL, USA) at a speed of 450 rpm. To ensure a uniform surface roughness, both sides of the samples were polished with a sequence of 400-, 600-, and 1200-grit SiC paper for 15 s at a constant speed of 100 rpm using a grinding machine (LabolPol-4; Struers, Madrid, Spain) under water cooling. To ensure a uniform thickness of the samples (±0.05 mm), we employed a precision digital caliper (Heavyware Tools) at three different points. The samples were then randomly assigned to the following experimental groups according to the resin-based luting agent and its shade (n = 10): two resin cements Variolink Esthetic in Light, Neutral, and Warm shades (Ivoclar Vivadent, Schaan, Liechtenstein) and RelyX Veneer, in B0.5/white, Translucent and A3

Opaque/yellow opaque shades (3M Oral Care, Seefeld, Germany); a flowable composite resin (G-aenial Flo in A2 and A3 shades (GC Europe, Leuven, Belgium); and a preheated composite resin (Filtek Supreme XTE, A3 Body shade (3M Oral Care, St. Paul, MN, USA)). The latter composite resin was used for the control group.

**Table 1.** Manufacturer and composition of ceramic, resin-based material and composite resin tested.

| Material and Manufacturer | Composition | Batch Number |
|---|---|---|
| Cerec® Blocs C/PC VITA Shade: A2 CAD-CAM feldspathic ceramic | $SiO_2$ (56–64%), $Al_2O_3$ (20–23%), $Na_2O$ (6–9%), $K_2$ (6–8%), CaO (0.3–0.8%), $TiO_2$ (0.0–0.1%), pigments <0.1%. | 66301 |
| RelyX Veneer 3M Oral Care Shade: B0.5, A3 and Translucent Resin cement | Bis-GMA, TEGDMA, Zirconia/silica, modified silica. Particle loading approximately 66% by weight, particle size approximately 0.6 mm, photoinitiator. | N862421 N816236 N843828 |
| Variolink Esthetic LCIvoclar Vivadent Shade: Light, Neutral and Warm Resin cement | Dimethacrylate, methacrylate monomers, inorganic particles Ytterbium trifluoride and spheroid oxide mixed. primers, stabilizers and pigments. Particle size is from 0.04 to 0.2 μm. Inorganic charge is approximately 38%. | v48653 w05218 w06171 |
| G-aenial Universal GC Corporation Shade: Flo A2 and A3Flowable composite resin | Urethanedimetrylate, Bis-MEPP, TEGDMA (31%). Silicon dioxide (16 nm), Strontium glass (200 nm), pigments (69%), photoinitiator. | 161202 |
| Filtek Supreme XTE3M Oral Care Shade: A3 Body Nanofilled composite resin | UDMA, Bis-GMA, Bis-EMA, Silica (20 nm) Zirconia (4–10 nm). Size of the particles together 0.6 to 10 μm. Inorganic particles represent 72.5% of the total charge. | N859611 |

Composite resin discs (Filtek Supreme XTE (3M Oral Care, St. Paul, MN, USA,)) (n = 180) with a thickness of 1 mm (±0.05 mm) were used as substrate. The composite discs were prepared using a resin former (sample ref. 7015 Smile Line Porcelain, St-Imier, Switzerland) and light-cured with a LED unit (Elipar S10; 3M Oral Care, Seefeld, Germany) for 40 s at high intensity (1000 mW/cm$^2$) according to the manufacturer's instructions. Resin discs were also calibrated using a digital caliper (Heavyware Tools). The ceramic samples were randomly paired with the resin disks to make 18 groups with 10 samples per group.

Surface treatment of the ceramic was carried out according to the manufacturer's instructions. Firstly, application of 9.6% hydrofluoric acid (PulpDent Corporation, Watertown, MA, USA) for 90 s, then rinsed for 60 s and followed by application of 37% phosphoric acid (R&S Supraetch; R&S, Paris, France) making vigorous circular movements for 60 s and using a microbrush. The ceramic samples were washed with distilled water, followed by an ultrasonic bath for 4 min. The surfaces were dried with 96% alcohol, and a silane coupling agent (Ultradent, South Jordan, UT, USA) was applied for 20 s and evaporated for 60 s. Finally, an adhesive system (Optibond™ FL; Kerr Dental, Orange, CA, USA) was applied without curing.

Each ceramic sample was cemented maintaining a constant force of 50 Newtons for 60 s [32] to standardize the luting agent thickness. In the control group, the composite resin used for luting (Filtek Supreme XTE, A3 Body shade) was previously heated in a resin oven (55 °C) (Micerium, Avegno, Italy) before its application.

Photopolymerization was carried out with the same LED unit for 40 s in the center of each sample. The intensity of the light was checked regularly with a Demetron radiometer (Kerr). After polymerization, the bonded samples were stored for 24 h in a dry environment and protected from light.

Fluorescence spectra of each sample was obtained on a spectrofluorometer (SPEX Fluorolog 2I2I; Horiba, Kyoto, Japan) at a wavelength of 380 nm and at room temperature. The area under each curve was integrated and used as a reference for each sample. For each group, a single spectrum was averaged.

The results of the fluorescence were statistically analyzed by a two-way ANOVA was performed to analyze the effect of the resin-based luting agent and the thickness of feldspathic ceramic (0.5 or 0.8 mm). Post-hoc comparisons were performed using Tukey and the Student's *t*-tests. All statistical tests were performed with a statistical software program (IBM SPSS v22; IBM Corp., New York, NY, USA) ($\alpha = 0.05$).

## 3. Results

The fluorescence spectrum of all tested materials showed similar pattern with a fluorescence peak around 450 nm and slowly decreased to 700 nm. In Figures 1 and 2 each color represents the average group pattern. Lower fluorescence emission intensity peaks were detected around 542 nm. The resin-based materials tested had different intensities of fluorescence emission.

Table 2 and Figure 3 show the mean fluorescence values (standard deviation, SD) for each experimental group. The two-way ANOVA revealed that fluorescence values were significantly influenced by the resin-based agent tested ($p < 0.001$), the ceramic thickness ($p < 0.001$), and the interaction between these factors ($p < 0.001$).

For feldspathic ceramic thickness of 0.5 mm (Table 2 and Figure 1), samples luting with flowable composite resin G-aenial Flo A3 obtained the highest fluorescence values although statistically similar to those luting with the same brand in A2 shade, Variolink Neutral and with the group cemented with preheated composite (reference group). The samples cemented with Variolink Esthetic, regardless the shade, obtained similar values of fluorescence and also similar to reference group. The lowest values of fluorescence were obtained for the three groups cemented with RelyX Veneer, regardless of the shade.

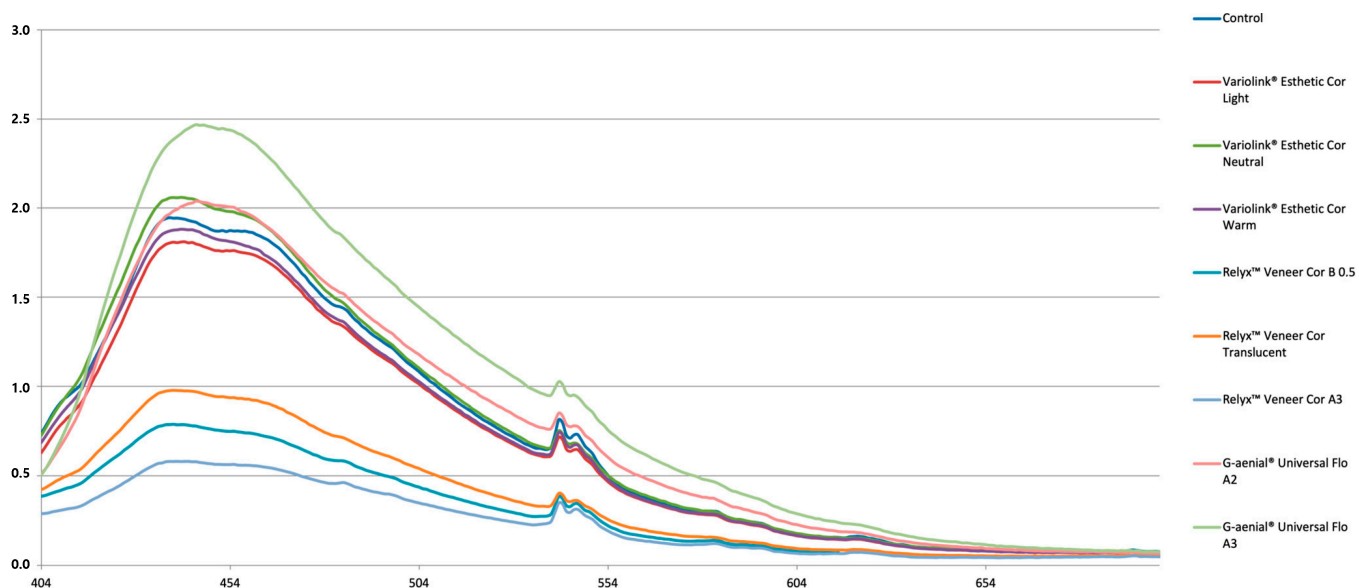

**Figure 1.** Fluorescence emission spectra different groups according to resin-based material used for luting feldspathic veneers (0.5 mm thickness). Graph values are in millions of a.u.

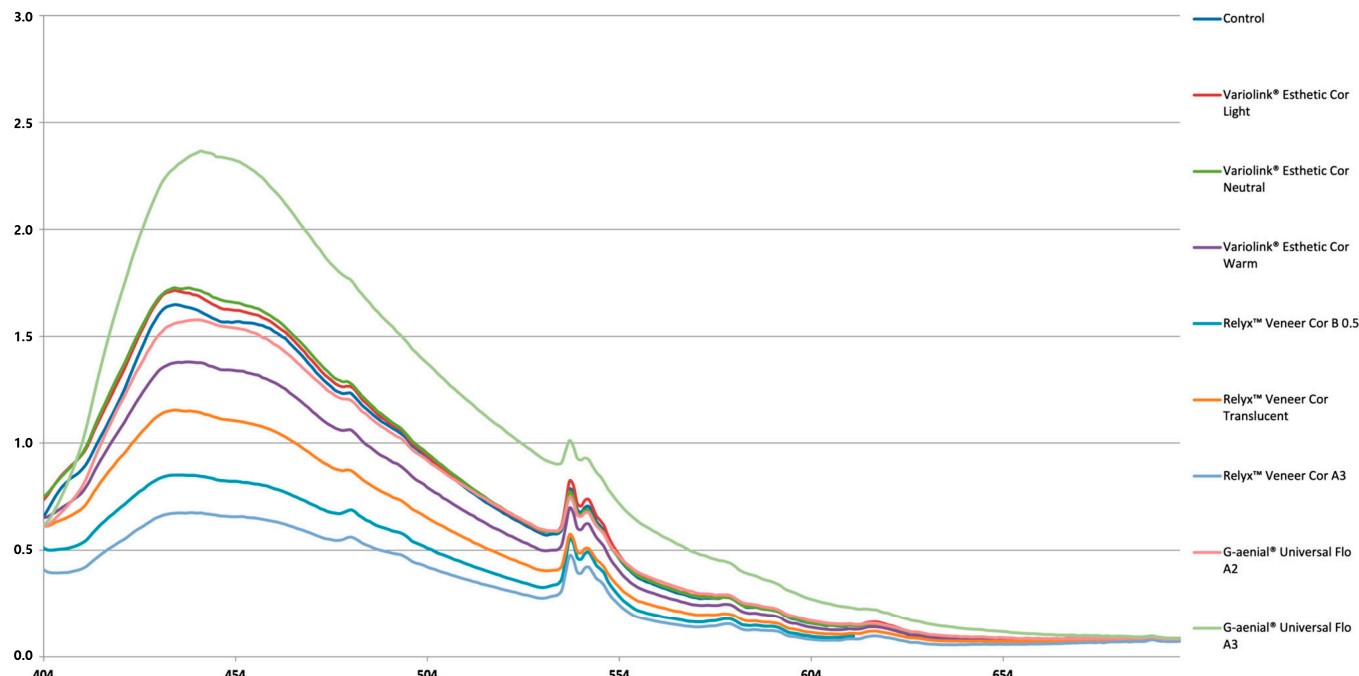

**Figure 2.** Fluorescence emission spectra different groups according to resin-based material used for luting feldspathic veneers (0.8 mm thickness). Graph values are in millions of a.u.

For the thickness of 0.8 mm (Table 2 and Figure 2), the highest fluorescence level was obtained for G-aenial Flo A3 group, followed by Variolink Esthetic Light and Neutral, G-aenial Flo A2, and the reference groups. The fluorescence of Variolink Warm group were lower than the groups luted with Light and Neutral from the same brand, and statistically similar to G-aenial Flo A2, RelyX Translucent, and the reference group. RelyX Veneer B0.5 and A3 groups yielded the lowest values, demonstrating that the latter group showed statistically lower fluorescence than RelyX Translucent.

Student's *t*-test did not show significant differences in fluorescence intensities between 0.5 and 0.8 mm thickness, except for Variolink Neutral and Warm, and G-aenial Flo A2 groups, in which the values decreased with veneer thickness of 0.8 mm.

**Table 2.** Mean fluorescence values (arbitrary unit, a.u.) and standard deviation (SD) obtained for bonded ceramic samples according to the resin-based luting agent and the feldspathic ceramic thickness (n = 10). For each column, different letters indicate significantly different fluorescence mean values among luting agents used for each feldspathic ceramic thickness.

| Resin-Based Material | Ceramic Thickness | | 0.5 mm vs. 0.8 mm (*p*-Value) |
| --- | --- | --- | --- |
| | 0.5 mm Mean ± SD | 0.8 mm Mean ± SD | |
| Variolink Light | $1.81 \times 10^6 \pm (3 \times 10^5)$ B | $1.71 \times 10^6 \pm (1 \times 10^5)$ E | 0.361 |
| Variolink Neutral | $2.06 \times 10^6 \pm (2 \times 10^5)$ BC | $1.73 \times 10^6 \pm (1 \times 10^5)$ E | 0.005 |
| Variolink Warm | $1.88 \times 10^6 \pm (2 \times 10^5)$ B | $1.38 \times 10^6 \pm (1 \times 10^5)$ CD | <0.001 |
| RelyX Veneer B0.5 | $8.01 \times 10^5 \pm (2 \times 10^5)$ A | $8.99 \times 10^5 \pm (1 \times 10^5)$ AB | 0.296 |
| RelyX Veneer Translucent | $9.78 \times 10^5 \pm (2 \times 10^5)$ A | $1.15 \times 10^6 \pm (1 \times 10^5)$ BC | 0.072 |
| RelyX Veneer A3 | $6.53 \times 10^5 \pm (4 \times 10^5)$ A | $7.78 \times 10^5 \pm (1 \times 10^5)$ A | 0.442 |
| G-aenial Flo A2 | $2.04 \times 10^6 \pm (3 \times 10^5)$ AC | $1.58 \times 10^6 \pm (2 \times 10^5)$ DE | 0.003 |
| G-aenial Flo A3 | $2.47 \times 10^6 \pm (6 \times 10^5)$ C | $2.36 \times 10^6 \pm (2 \times 10^5)$ F | 0.630 |
| F Supreme XTE A3 (preheated) | $2.02 \times 10^6 \pm (2 \times 10^5)$ BC | $1.65 \times 10^6 \pm (2 \times 10^5)$ DE | 0.05 |

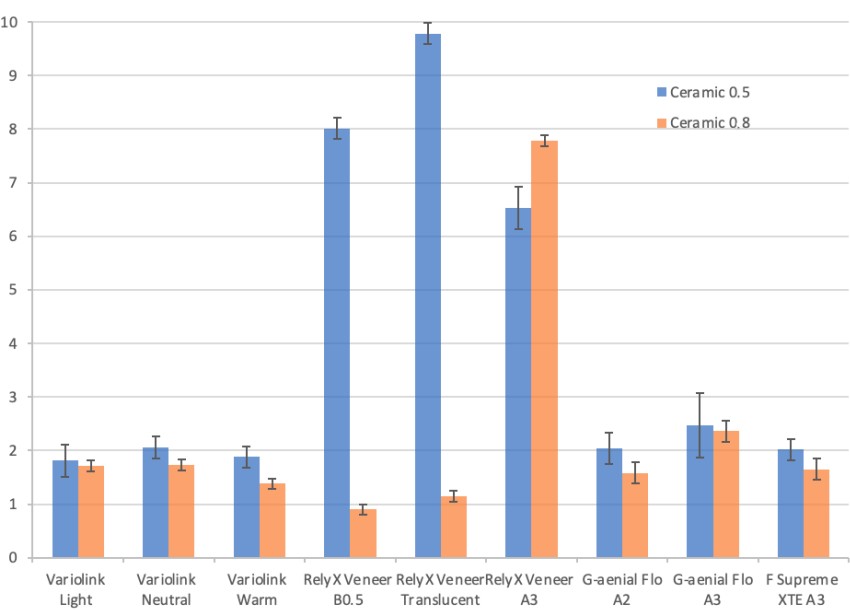

**Figure 3.** Mean fluorescence values (millions of arbitrary unit, a.u.) and standard deviation bars obtained for bonded ceramic samples according to the resin-based luting agent and the feldspathic ceramic thickness (n = 10).

## 4. Discussion

According to the results of the present study, the research hypothesis was rejected because the emission intensity of fluorescence of feldspathic ceramic restorations is influenced by the shade of resin-based material, as well as by the thickness of feldspathic veneer.

In order to achieve a natural looking restoration, the restorative materials should imitate the optical properties of the natural tooth, including fluorescence [15]. The wavelength of the excitation beam was 360 nm, which is the wavelength that causes a peak emission fluorescence intensity of the tooth. Ceramic veneers are usually fabricated as thin layers; therefore, the type and color of the cementing agent could influence the fluorescence of the restoration, as well as the final result [27].

Different levels of fluorescence were detected among groups tested. It is evident that the high translucency of the feldspathic ceramic allows the fluorescence of the luting agent to influence the final fluorescence of the restoration. Hence, the fluorescence values of restorations are the result of combination of the fluorescence of feldspathic ceramic and the underlying cement filtered by the ceramic [32].

In the present study, only one type of ceramic has been evaluated. Although fluorescence of dental ceramics have been previously reported [18,21,27,32–35], literature about this optical property in current ceramics is scarce [21,27] and, as far as we know, there are no data about the ceramic (CEREC Blocs) used in this study. Feldspathic ceramic is characterized as an extremely aesthetic material, indicated to mimic the dental structure [36]. The ceramic used in this study was a CAD-CAM feldspathic ceramic, available for digital manufacturing, and sintered by optimized industrial procedures, which results in blocks with fewer flaws and pores, and better mechanical properties than the traditional or hand-built ceramic [37,38].

Basic components of ceramics and dental resins are not able to produce fluorescence, so this can be achieved by incorporating rare-earth oxides—such as europium, terbium, and cerium—which have a strong fluorescence when exposed to UV light [17,19,33,39]. Other procedures, such as the application of an external fluorescent glaze layer on a pressed lithium disilicate ceramic, have been recently reported [21]. Regarding composite resin, manufacturers do not disclose the exact composition of these materials, although it is known that some luminescent species—such as rare-earth oxides, terbium coordination polymers of PEMA, or aromatic complex—have frequently been used [19,22,24].

Despite the fact that fluorescence is one of the optical properties of natural teeth that has attracted the attention of dental professionals in recent years [19], papers on this subject are limited. Silami et al. [28] also quantified the fluorescence of two different ceramic veneers (lithium disilicate glass-ceramic and fluorapatite glass-ceramic) and two types of resin cements (light-curing versus self-adhesive resin cements). However, the difference in materials and methodology does not allow a direct comparison of the results.

With ceramic veneers of 0.5 mm thickness, the groups in which Variolink Veneer and the flowable composite resin G-aenial were used, exhibited similar fluorescence values to the group used as reference (preheated Filtek Supreme XTE). In contrast, the fluorescence intensities were the lowest when RelyX Veener resin cements were used. These results might be explained by the luminophore content of the cements evaluated. In the case of Variolink Veneer, it contains ytterbium trifluoride, a compound that provides fluorescence [40]. It was previously reported that the light-cured resin cement Variolink II (Ivoclar Vivadent), also includes ytterbium trifluoride, and improved the fluorescence level of the e.max Press ceramic restoration [28]. On the other hand, the manufacturer of RelyX Veneer cement does not report luminophores in its composition. This may possibly explain why the fluorescence level was lower than in the other groups.

In the groups with ceramic thickness of 0.8 mm, a greater influence of the shade of the cement was appreciated between cements of the same brand. Darker shades within the same brand obtained a lower emission of the final fluorescence of the restoration. Thus, for example, veneers cemented with RelyX Veneer A3 had lower fluorescence values than RelyX Veneer Translucent. The same was observed with G-aenial Flo A3 compared to A2, and Variolink Veneer Warm compared to Light and Neutral. This trend was also confirmed in a recent paper in which the authors investigated the fluorescence behavior of different shades of selected contemporary tooth-colored restorative materials and concluded that— within any one brand of material—fluorescence emissions differed according to shade, with the lightest shades giving the strongest emissions [41].

According to the results, the fluorescence values emitted with thicker veneers showed a decrease with respect to 0.5 mm thickness significant for G-aenial A2, and Variolink Neutral and Warm groups. This decrease can be due to the fluorescence emitted mainly by the ceramic, since the luting material does not have constituents with the capacity to overcome the emission of fluorescence by the ceramic itself. RelyX Veneer cement, regardless of its shade, maintained the lowest fluorescence values.

In the present study, the composite resin Filtek Supreme XTE was used as a substrate instead natural tooth in order to avoid biological variability [27]. It is known that fluorescence in natural teeth is a multifactorial phenomenon based on multiple organic and inorganic components, age, and biotype [19,42]. Furthermore, fluorescence is lost after extraction unless fixation procedures are performed [42]. Thus, to replicate this optical property artificially, Filtek Supreme XTE was selected due to previous papers revealing an optimal fluorescence similar to the natural tooth [42–44].

Fluorescence makes the teeth look brighter and whiter in daylight [45]. Therefore, fluorescence appears in UV light above all, but ambient light is also relevant, because it influences the color of restorations. In our study, UV light has been used, but ambient light can also induce a certain degree of fluorescence, and that is why it has been evaluated in different studies [17,18,27,32–35].

The peak of fluorescence emission intensity was determined to be around 450 nm. In the visible light spectrum, this wavelength corresponds to the blue color. The blue complement given by the fluorescence present in the cements should be taken into account not only by the restorative dentist, but also by the laboratory technician. Thus, the communication between the dental office and the laboratory should also indicate the type of cement that is going to be used.

Natural teeth have fluorescence intensity peaks that are located at wavelengths of 350, 360, 405, 410, and 440 nm [46], which is in agreement with the obtained results, where the materials under this study present two peaks of fluorescence emission intensity around

450 nm and at 542 nm, and according to previous reports [22,47–50]. This indicates that the materials used in the present study may contain similar elements with fluorescence emission capacity, but in different percentages.

Limitations of the present study include the use of only one ceramic and specific brands of resin cement, so the behavior may vary with other materials. Further studies are recommended to assess the change of the emission of fluorescence with other ceramics and resin cements.

**5. Conclusions**

Based on the results obtained in this in vitro investigation, it is possible to conclude that:

- The fluorescence of feldspathic ceramic veneer restorations (CEREC Blocs) can be influenced by the shade and brand of resin-based materials used for luting;
- Thicker feldspathic veneers show less fluorescence emission intensity when they are cemented with resin cements or flowable composite resins in darker shades.

**Author Contributions:** Conceptualization, J.s.d.C.P. and J.A.R.; Data curation, F.M.; Formal analysis, F.M.; Investigation, J.S.d.C.P.; Methodology, J.A.R.; Project administration, P.M. and M.V.F.; Resources, J.S.d.C.P.; Software, F.M.; Supervision, P.M. and M.V.F.; Validation, J.A.R., F.M., P.M. and M.V.F.; Visualization, J.A.R.; Writing—original draft, J.S.d.C.P.; Writing—review and editing, J.A.R. and M.V.F. All authors have read and agreed to the published version of the manuscript.

**Funding:** This research received no external funding.

**Institutional Review Board Statement:** Not applicable.

**Informed Consent Statement:** Not applicable.

**Data Availability Statement:** The data presented in this study are available on request from the corresponding author.

**Conflicts of Interest:** The authors declare no conflict of interest.

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
