# Peer review of "The Effect of Feldspathic Thickness on Fluorescence of a Variety of Resin Cements and Flowable Composites"

_applsci, doi:10.3390/app12136535_

Round 1

Reviewer 1 Report

This manuscript investigated the effects of luting cement and ceramic thickness on the fluorescence of ceramic. Overall, this study was well designed and conducted. However, there are some minor aspects could be improved:

1.       What is the rationale to use 0.5 and 0.8mm thickness of the ceramic?

2.       Pleas provide high resolution figures of figure 1 and figure 2.

3.       It might be easier to understand the results for reader if the results are presented in a bar graph.

Author Response

  1. What is the rationale to use 0.5 and 0.8mm thickness of the ceramic?

Our choice was made in a clinical sense. These two widths are normal for a veneer and we assume the cement material would change the fluorescence the most.

  1. Please provide high resolution figures of figure 1 and figure 2.

Done.

  1. It might be easier to understand the results for reader if the results are presented in a bar graph.

Done.

Reviewer 2 Report

Dear Authors

The topic of the Ms is interesting. The fluorescence of resin-based luting  material is crucial to achieve the best esthetic result. Although the Ms is well structured, more details are needed in Introduction and Material and Methods section. For these and other points I suggest a minor revision. For comments, please see the attached file. 

Author Response

All English typos were corrected

Table 1 has already a legend, we don’t understand what else the reviewer wants.

Micerium is the resin oven not a composite used in the study therefore not present on table 1. The control group is Filtek Supreme XTE

The initial fluorescence of the Composite resin discs and of CEREC Blocs are not presented has it’s not our goal with study. But we understand the point made by the reviewer.

The comment referring to the negative group, we think the reviewer is talking about the control group. The control group is the line with the lowest value

Table 2 comments asks for “Please, specify the meaning of "A,B and C”. The legend for this table reads “For each column, different letters indicate significantly different fluorescence mean values among luting agents used for each feldspathic ceramic thickness.” Does the reviewer want further explanation?

G-aenial Flo does not have a “w”

We opted out of discussing the traditional or even new protocol for cementation has it was not our goal in this study.
